# Participation in Low Back Pain Management: It Is Time for the To-Be Scenarios in Digital Public Health

**DOI:** 10.3390/ijerph19137805

**Published:** 2022-06-25

**Authors:** Michela Franchini, Massimiliano Salvatori, Francesca Denoth, Sabrina Molinaro, Stefania Pieroni

**Affiliations:** Institute of Clinical Physiology, National Research Council, 56124 Pisa, Italy; msalvatori@ifc.cnr.it (M.S.); francesca.denoth@ifc.cnr.it (F.D.); sabrina.molinaro@ifc.cnr.it (S.M.); stefania.pieroni@ifc.cnr.it (S.P.)

**Keywords:** participation, digital health, Dress-KINESIS, scenario method, Arena tool, low back pain, modeling

## Abstract

Low back pain (LBP) carries a high risk of chronicization and disability, greatly impacting the overall demand for care and costs, and its treatment is at risk of scarce adherence. This work introduces a new scenario based on the use of a mobile health tool, the Dress-KINESIS, to support the traditional rehabilitation approach. The tool proposes targeted self-manageable exercise plans for improving pain and disability, but it also monitors their efficacy. Since LBP prevention is the key strategy, the tool also collects real-patient syndromic information, shares valid educational messages and fosters self-determined motivation to exercise. Our analysis is based on a comparison of the performance of the traditional rehabilitation process for non-specific LBP patients and some different scenarios, designed by including the Dress-KINESIS’s support in the original process. The results of the simulations show that the integrated approach leads to a better capacity for taking on patients while maintaining the same physiotherapists’ effort and costs, and it decreases healthcare costs during the two years following LBP onset. These findings suggest that the healthcare system should shift the paradigm towards citizens’ participation and the digital support, with the aim of improving its efficiency and citizens’ quality of life.

## 1. Introduction

Prospective planning of financial resources for facing chronicity is a complex and intersectoral task, as it involves both healthcare and social support systems. Furthermore, it is particularly challenging because it concerns the mismatch between complex community needs and current health/social supply [1].

Strategic decision-making processes in the fields of healthcare and public health are characterized by high levels of uncertainty, innovation and change. Simulation modeling is a complex foresight method that is particularly helpful for strategic planning, as it makes use of both qualitative (e.g., expert opinion and discussion) and quantitative elements (e.g., scenario calculations). Systematic use of the scenarios method started with military planning in the 1960s, but many scenario projects were developed after the year 2000 to address a wide range of topics in the fields of public health and health policy research [2].

The creation of scenarios needs an interdisciplinary approach to foster systematic and structured discussion of possible future alternatives by the incorporation of expert knowledge. Additionally, the scenario method requires a thorough fact-gathering process similar to a SWOT, SOAR or Delphi process, but distinctly from a SWOT analysis, it considers and weighs the forces and uncertainties that will influence the future. 

The scientific literature reports that the quality of the scenarios depends on the information basis and the competency of the experts taking part in the building process [3]. ‘Ordinary’ people acting as experts may be included in the scenario planning [4], in agreement with the framework of participatory strategies. This implies considering the patients’ lived experience as a source of evidence on a par with the clinicians’ experience [5]. Participation in health concerns the involvement of the citizens in data collection and the use of electronic health or mobile health (mHealth) care apps to promote the self-management of chronic diseases. These tools should be designed according to the users’ lived experience and validated by the users themselves [6]. Furthermore, a number of articles suggests that care providers “meet patients where they are” by combining strategies to improve health literacy with the Chronic Care Model approach [7].

Inviting patients to participate in their own care could involve facing the critical issues of misinformation, disinformation and information filtering on the collective level, which impact the common knowledge and perception of the real efficacy of care [8].

As an example, for musculoskeletal chronic pain management, the most popular social media platforms currently propose many alternative therapies (telerehabilitation plans, dietary supplements, herbal or synthetic drugs, acupuncture, chiropractic, etc.) leveraging the patient’s dissatisfaction with conventional treatments or their lack of time to adhere to such treatments. Self-medication, one-size-fits-all therapy and non-expert consultation are potentially harmful behaviors, in particular for patients presenting comorbidities [9].

The use of digital health tools is a new method to meet patients’ preferences, priorities and goals [10] and to improve their empowerment and self-education [10]. This is especially the case at the present time: both the “one health” approach, promoted by the European Commission, and the National Recovery and Resilience Plan (NRRP) suggest improving the use of digital data and technology. They encourage the sharing and use of eData and big data, the e-Care and Connected Care projects and the evolution of the electronic health record approach towards citizens’ files [11].

The definition of digital public health as an asset that the public health community can use to fulfill its aims and mission to ensure quality, accessibility, efficiency and equity of healthcare [12,13] leads to the conceptualization of the digital transformation as a complex and multifaceted process [14]. Furthermore, the digital transformation in some of its aspects represents a fundamental change in the mode and culture of care delivery [14]. All of this makes the impact of digital health technologies liable to be best assessed not only from a technological point of view. The issues in terms of regulations (compliance with the General Data Protection Regulation), methodology (AI solution safety, explicability and fairness), organization both at the health professional and societal levels (easy-to-use healthcare solutions, digital and health literacy, equality and inclusivity), cost and resource savings and health outcomes should be considered as well [15].

Mobile health services are an example of digital health services that already impact the process of healthcare delivery by making it more accessible, affordable and available. As the World Economic Forum (WEF) emphasizes, digital health services should focus on continuous monitoring, connected homes, intelligent treatments and virtual care teams by involving important shifts from diagnosis and treatment to disease prevention and management [16].

Presently, there are over 300,000 health applications on the market, and that number continues to grow. Approximately 40% of smartphone owners use mHealth apps. Due to the current COVID-19 pandemic situation, the existence of mHealth has become much more important, but more than 30% of mHealth applications are uninstalled within a month of being downloaded [17]. 

One cause of discontinuation of use of these tools is the limited and lacking theoretical basis of these systems, which are not always based on the research literature [18]. 

Among chronic conditions, low back pain (LBP) is the leading cause of years lived with disability and has a high economic burden in terms of direct costs (medications, hospitalizations and outpatients), indirect costs (loss of productivity, disability-adjusted life years) and intangible costs (reduced enjoyment of life, including the inability to interact with others) [19]. 

The PACO (PAin COsts) study, involving 52 general practitioners (GPs) and 581 patients in Italy (mean age 62.4 years, 71.8% female; 52.0% dorsopathies as pain-related diagnosis, mean number of years with pain 7.7), estimated chronic musculoskeletal pain-related healthcare costs in terms of specialist consultations, diagnostic and laboratory tests, hospital admissions, drug use and GP consultations [20]. Average costs per patient per year amounted to EUR 212.6 (95% CI: EUR 180.0–255.3), with more than two-thirds of the patients (70.6%) treated with non-selective NSAID-based therapy and/or cyclo-oxygenase-2 (COX-2) inhibitor-based therapy. The highest costs concerned hospital admissions (EUR 52.3; 95% CI: EUR 25.6–90.3), GP consultations (EUR 51.4; 95% CI: EUR 48.7–54.1), drugs for pain relief (EUR 37.3; 95% CI: EUR 32.1–43.2) and physiotherapy (EUR 31.9; 95% CI: EUR 26.1–38.5) [20]. Patients treated with pain-relieving drugs had an average cost per year four times greater than that of untreated cases (*p* < 0.01).

Notwithstanding the major international clinical guidelines prioritizing non-medical approaches for LBP patients, most health systems are not well equipped to support tailored physical therapies, and/or patients are unable to access rehabilitation centers (e.g., living in rural/remote areas, care costs, lack of time, etc.) [21].

Some evidence [22,23] showed that almost half of LBP cases managed within general practice had no access to the recommended multidisciplinary approach to pain management, particularly when LBP was uncomplicated.

With the aim of shifting the paradigm for LBP management and costs, we present a new possible method based on an mHealth tool, the Dress-KINESIS, designed and implemented by a group of researchers with different skills and areas of expertise (epidemiologists, kinesiologists, physical therapists and IT developers). The tool is designed for: (a) collecting real patient syndromic information about LBP and its determinants; (b) proposing targeted self-manageable plans of exercise for improving pain and disability; (c) identifying and validating the tailoring criteria to qualify the most efficient exercise protocols for each profile of limitation; (d) promoting the role of physical activity as a therapeutic strategy in reducing pain and limitations from LBP; (e) sharing valid educational messages to support prevention; and (f) monitoring the efficacy of tailored workouts in terms of health outcomes and adherence.

The Dress-KINESIS has been designed to support the maintenance phase of the exercise protocols of patients with a good prognosis after they have performed a first cycle of supervised activity within a rehabilitation institute (expressed needs). Additionally, it aims to engage citizens who did not seek care (non-expressed needs) in managing all of the phases of exercise protocols for LBP relief (first assessment, protocol development, protocol execution, prognosis evaluation and maintenance activity). A high level view of Dress-KINESIS’ design is shown in Figure 1.

We developed a simulation model to compare the behavior and outcomes of traditional rehabilitation processes for non-specific LBP patients, performed within a theoretical model of a rehabilitation institute (as-is model), with different scenarios (to-be models) that include Dress-KINESIS support in the original process. This paper describes our simulation model and the results of its application.

## 2. Materials and Methods

### 2.1. The Dress-KINESIS Tool

The Dress-KINESIS, already described in Franchini et al., 2021 [1], is a customization of the Dress system (“Doing Risk sElf-assessment and Social health Support”), an mHealth system based on the Telegram bot and developed during the recent COVID-19 pandemic [24]. 

The DRESS-Kinesis aims to collect the most exhaustive data about physical exercise habits, LBP severity status, treatments and costs concerning the subpopulation of users declaring to have experienced LBP or lumbar tightness (Figure 2). The tool collects data by asking users a set of questions (Figure 2) grouped into clusters of 7–10 daily questions for improving responsiveness and promoting a long-lasting collaboration with users. Survey questions have been formulated by reviewing the most relevant scientific references about LBP assessment and management and referring to the biopsychosocial model for investigating the behavioral, psychological and social factors associated with LBP. Additionally, the Dress-KINESIS shares with the users many evidence-based educational messages about LBP prevention on the basis of users’ answers to some specific items (see the items flagged with the red asterisk in Figure 2). For each message, the bibliographical reference is reported.

Information collected about each user allows us to characterize his LBP severity level at the baseline (T0) and to group subjects into three levels of limitation. Groups A, B and C are identified according to the user’s abilities in coping with daily living activities (Oswestry score) [25] and some other abilities, identified through the answers provided to specific items of the Low Back and Lower Extremities (LEAFS) scale [26].

The Dress-KINESIS logical scheme refers to the treatment-based classification (TBC) system for patients with LBP [27]. The “movement control approach” is proposed for subjects with moderate pain and disability status, as the goal of the treatment is improving muscle impairments to allow performance of activities of daily living. The “function optimization approach” is proposed for subjects with low pain and disability status, as the goal of the treatment is to improve the patient’s ability to perform higher levels of physical function without symptom exacerbation.

Subjects with severe pain and disability status (Oswestry score > 40%) are excluded from Dress-KINESIS use (Figure 3). Comorbidities such as cardiovascular diseases, spine injuries, neurological conditions, inflammatory diseases, suspected/known cancer, pregnancy or any clinical ban on moderate exercise are considered as further contraindications to participate in self-management exercise programs. To ensure the safety of users showing contraindications, the Dress-KINESIS suggests that they undergo a thorough clinical evaluation before performing any type of physical exercise. According to the hypothesis that patients who presented with acute or persistent LBP improved markedly in the first 6 weeks of treatment [28], users who do not show contraindications and state their willingness to follow the tailored workouts are invited to participate in a 9-week tailored exercise program, divided into three 3-week cycles (Figure 3).

Each program consists of two sections of daily activity, to be performed for 3 days a week:The WO section (lasting from 45 min to one hour) includes a warm-up phase and some daily exercises aimed at strengthening the core muscles and the deep abdominal muscles, promoting respiratory control and increasing the range of motion of the spine and joints as well as overall body flexibility. These movements focus on improving balance, developing the muscular rate of force and reducing the neuromuscular fatigability.The AE section (lasting about 30 min) includes some specific aerobic activities (running, walking and climbing steps) and stretching movements. The AE section aims at preventing/limiting bone loss and metabolic syndrome. The AE sections are tailored based on some abilities of the tool users, identified through the answers provided to specific items of the LEAFS scale.

We classified the exercises to be included in each WO section by considering the following criteria:Aim (improving balance, strengthening the core muscles, hip mobilization, spine flexibility, promoting respiratory control, etc.);Starting position (lying prone or supine, standing, kneeling, etc.);Type of movement;Target muscles;Effort intensity, based on the work metabolic rate of each type of exercise (MET, metabolic equivalent) [29] and the number of repetitions planned for each user.

This system of classification allows us to identify groups of exercises with different effort intensities in order to personalize the individual workouts by level of limitation and between different sections of exercise. Each group includes some exercises with the same aim and target muscles, but with different starting positions and/or types of movement.

Before starting with the first 3-week cycle, each user performs 1 week of exercise feasibility testing to plan his workout sessions and so promote adherence to the self-management plans (Figure 4).

Adherence to the exercise plans is assessed through the ratio between the number of exercises actually carried out and those proposed within each cycle.

During the 1-week exercise feasibility testing, each user will receive all of the exercises, explained in video format. This allows us not only to guide him in the required movements but also to highlight the postures/positions to be kept under control to ensure the correct execution of the exercise. While performing each one, the user will have to: (a) identify his own directional preference (DP: situation in which movement in one direction improves pain and limitation of range of motion) and (b) count the maximum number of repetitions (MNR) that he can perform while maintaining the correct position. The tool will collect the DP and MNR data to further customize the WO sections (Figure 3).

Each WO section increases in intensity (type of exercise) and volume (number of repetitions) cycle by cycle. As shown in Figure 4, all users (groups A, B and C) after having completed the first 3-week exercise cycle, will fill in the LEAFS scale (LEAFS T1) once again and fill in it for the third time after 6 weeks of activity (LEAFS T2).

The users in group A progress to the subsequent cycles of activity following the time criterion only (3 weeks for each cycle). The users in the groups B and C will be evaluated to decide if they can progress with new plans of exercise at greater intensity and volume. Progression criteria (criteria n.2–5) refers to the minimum clinically important difference (MCID) for the LEAFS scale. The group B and C users who show a gap between LEAFS T0 and LEAFS T1 discordant from criterion n.2 will repeat the same WO sections of the first 3-week cycle of exercise. Conversely, they will progress from the first to the second cycle if the criterion is met. The transition from the second to the third cycle of activity (final 3 weeks) will happen if the gap between the LEAFS T2 and the LEAFS T1 scores meets criterion n.2. At the end of the third exercise cycle, all of the users will fill in the LEAFS (LEAF T3) and the Oswestry (Oswestry T1) scales again. In each cycle transition, all of the users (groups A, B and C) who meet the criterion n.1 are invited to stop the self-management programs and to undergo an in-person clinical assessment, under the hypothesis that their lumbar pain depends on a specific undiagnosed condition/disease (Figure 4). Additionally, the users in group B or C will be stopped if, after 6 weeks of activity within the same cycle, they do not meet the criterion of progression n.2. The progression criteria are not clearly described in this document, as the procedure to protect the Dress-KINESIS’s intellectual property rights is still in progress.

### 2.2. The Scenarios Approach

A scenario can be defined as a description of a possible future state or condition within a subject field. The basic criterion for the inclusion of a scenario in a scenario set, therefore, is not the probability that it will eventually happen but the fact that it might happen given certain assumptions about the surrounding world. This makes the selection of the key factors a crucial point of a scenario development [30].

In the LBP field, the key factors to consider are the following:During 2017, the LBP age-standardized point prevalence was 7.50% (95% CI: 6.75–8.27%) [31] and the 1-year incidence of any episode of lumbago (acute episodes and recurrences) ranged from 1.5 to 36.0% [32].The Ministry of Health surveyed 1141 rehabilitation institutes in Italy. During 2019, each institute counted an average of 9.91 outpatient accesses per day (95% CI: 2.17–17.65%) in the area of motor rehabilitation (expressed needs) [33].About 90% of all patients show non-specific LBP [34]. To manage uncomplicated acute LBP, international guidelines recommend the provision of advice, education, reassurance and simple analgesics, but 33% of patients who follow first-line care experience a recurrence in the next 12 months, and 20% to 30% develop chronic pain [35].Early engagement of patients reduces healthcare costs through the avoidance of unnecessary investigations and treatments [36]. Childs et al. estimated that in the 2-year follow-up period after a new episode of LBP, patients involved in early physical therapy protocols (within 0–14 days following pain onset) save EUR 1106 of healthcare costs (prescription of drugs and hospital costs) compared to patients undergoing delayed physical therapy [37].The proportion of LBP patients who seek care in public or private rehabilitation centers is on average 20% [38], with a higher percentage among those with chronic LBP (about 80%) [39].Between 58% and 75% of LBP patients (medium and high risk of poor prognosis) require further physiotherapy treatments after having performed the first activity protocol [40].LBP therapy strategies mainly depend on pain classification in terms of intensity and duration [41]. Many clinical trials have been conducted to evaluate the effects of particular physical interventions, but heterogeneity among LBP patients included in the same intervention tends to dilute the treatment results [42].Subgroup-matched treatment approaches have been proposed to improve evidence-based guidelines for supporting clinical decision-making. In the field of conservative management of LBP, the Treatment-Based Classification (TBC) system was first described by Delitto and colleagues in 1995 and has been further updated based on emerging evidence [27].

Based on these key factors, we used the Arena simulation system to design and mimic the behavior of traditional rehabilitation exercise protocols for LBP patients performed within a theoretical model of a rehabilitation institute (as-is model). Arena is a Microsoft Windows ^®^ application for studying a wide variety of simulation models of real-world systems by numerical evaluation. It is built on the SIMAN language construct and other facilities and augmented by a visual front end [43].

Furthermore, we used the Arena tool to compare the as-is scenario with other simulations (to-be scenarios) built by modifying one or more parameters of the original system (as-is scenario) to simulate Dress-KINESIS use.

### 2.3. The As-Is Scenario

The overall as-is scenario is shown in Figure 5.

In the as-is logical scheme, we defined the subsequent process elements:Entities, the dynamic objects in the simulation, representing non-specific LBP patients;Resources including (a) four physiotherapists with their work cost (EUR 12.00/h) and (b) drug prescription and inpatient costs during the 24 months after LBP onset (EUR 1106.00). Healthcare costs regard patients who did not express their care needs only.

We used the different Arena flowchart modules and their setting parameters to describe all of the stages of the dynamic process in the as-is model (Figure 5 and Figure 6a–c):

Total Needs create module (Figure 5 and Figure 6a), set by considering: (a) total demand of 5.8 patients/day per rehabilitation institute, (b) time between arrivals of patients with expressed and non-expressed needs, following a “random(expo)” distribution with an average interarrival time of 0.98 h (λ) and an arrival rate of 1.02 patients/hour (1/λ). These parameters are estimated from an LBP point prevalence of 7.5% [31].Attribute module: TNOW. This attribute is set to the system time (TNOW) when a new patient arrives. It records the simulation clock time as the model progresses.Pain Intensity decide module (Figure 5 and Figure 6a): used for the creation of two different pathways of care demand based on the patient’s pain intensity (74.5% low/medium intensity; 25.5% high intensity) [41].Lower Pain Need decide module (Figure 5 and Figure 6a): used for the creation of two different pathways of needs among patients with lower levels of pain (16% expressed vs. 84% non-expressed) [38].Lower Pain delay module (Figure 5 and Figure 6a): used for modeling the list of patients with lower pain waiting to access their first evaluation by a physiotherapist. The waiting time follows a triangular distribution of 3, 5, 7 days.Higher Pain Need decide module (Figure 5 and Figure 6a): used for the creation of two different pathways of needs among patients with higher levels of pain (80% expressed vs. 20% non-expressed) [38].Higher Pain delay module (Figure 5 and Figure 6a): used for modeling the list of patients with higher levels of pain waiting to access their first evaluation by a physiotherapist. The waiting time follows a triangular distribution of 2, 3, 4 days.PhysioTime hold and signal module (Figure 5): used for restricting the physiotherapists’ working days to Monday-to-Friday intervals.ProtDevTime seize–delay–release module (Figure 5 and Figure 6b): used for modeling the time spent by a physiotherapist in developing the tailored exercise protocol of each patient. This time follows a triangular distribution of 20, 30, 40 min.ProtSupervTime seize–delay–release module (Figure 5 and Figure 6b): used for modeling the time spent by a physiotherapist on the supervision of each patient during protocol execution. Under the hypothesis that a physiotherapist spends three working hours per week on each patient and follows a group of three patients at a time, the physiotherapy daily effort is set to 2.5% of his total time. Additionally, the ProtSupervTime follows a triangular distribution of 20, 30, 40 days.Assessment0 (and Assessment1) seize–delay–release modules (Figure 5 and Figure 6b): used for modeling the time spent by a physiotherapist to perform the first clinical assessments and the control visits for each patient. This time follows a triangular distribution of 10, 15, 20 min.Prognosis decide module (Figure 5 and Figure 6b): used for differentiating patients based on their prognosis after the first cycle of activity (74% poor/medium prognosis; 26% good prognosis) [40].New Protocol seize–delay–release module (Figure 5 and Figure 6b): used for modeling the time spent by a physiotherapist on the supervision of each patient with a poor/medium prognosis during the adjunctive protocol execution. This module has the same parameters of the ProtSupervTime seize–delay–release module.OUTLow Prog and LongRunOUTCOME dispose modules (Figure 5 and Figure 6b) which represent patients with a poor/medium prognosis or a good prognosis who exit from the model at the end of the second supervised cycle of activity or after the maintenance protocol, respectively.

The flowchart and data modules in the as-is model are related to each other by the name of the shared «physiotherapists» resource.

To estimate the Dress-KINESIS tool support, the as-is general model also includes two sub-pathways to be activated in the to-be simulations only by leveraging the:(1)DKExpressed decide module (Figure 5 and Figure 6b), concerning patients with a good prognosis after the first cycle of activity whose maintenance activity protocols could be defined and managed using mHealth technology;(2)DKnotExpressed decide module (Figure 5 and Figure 6c), concerning patients who did not access the traditional rehabilitation process and who are potential carriers of healthcare costs in the 24 months after LBP onset. (NotExpress Cost 24 months seize–delay–release module (Figure 5 and Figure 6b).

### 2.4. The To-Be Scenarios

We designed the to-be models by considering two different care pathways that could be improved by Dress-KINESIS use (Figure 5 and Figure 6a–c).

Along the expressed needs pathway, the tool could help rehabilitation institutes in the maintenance phase of activity of patients with a good prognosis. Within Arena, this has been modeled by modifying the percentage of patients with a good prognosis passing through the traditional maintenance protocol of activity by acting on the DKExpressed decide module (set as 0.0% in the as-is model shown in Figure 5 and Figure 6b). Along the non-expressed needs pathway, the Dress-KINESIS could support patients in all phases of their rehabilitation treatment (first assessment, protocol development, protocol execution, prognosis evaluation and maintenance activity). Within Arena, this has been modeled by modifying the percentage of patients who access the Dress-KINESIS protocols (Figure 5 and Figure 6c) by acting on the DKnotExpressed decide module (set as 0.0% in the as-is model).

Patients who exit from the maintenance protocols (Figure 5 and Figure 6b) and patients who exit from the Dress-KINESIS protocols (Figure 5 and Figure 6c) are represented and counted by the two dispose modules Dress-KINESIS Support and NOT ExpressedNeeds.

### 2.5. Systems Replication Parameters

The as-is and the to-be simulations have been set up according to these parameters:
NUMBER OF REPLICATIONS (of each model): 100;STARTING DATE AND TIME of replication: Friday 2021/01/01 9:00;LENGTH OF EACH REPLICATION: 30 days;HOURS PER DAY of replication: 8.

### 2.6. Output Measures and Statistical Analysis

The aim of this work is to compare the overall performance of traditional exercise protocols for non-specific LBP patients performed within a theoretical model of a rehabilitation institute (as-is scenario) with the to-be scenarios designed by including Dress-KINESIS use. Scenarios have been compared in terms of: (a) average number of patients who enter the system, (b) average number of patients in the system at any given time (WIP), (c) average number of physiotherapists in use over time, (d) average number of waiting patients and time spent before seeing a physiotherapist, (e) physiotherapists’ cost and (f) healthcare costs for patients who did not seek care within 14 days of need onset (non-expressed needs).

Once each scenario was simulated 100 times, the Arena tool produced a document (Category Overview Report) reporting the average and the 95% confidence interval’s half-width values for each output measure, as shown in Table 1.

Within Arena, statistics are actually random variables based on the results of the total number of replications of the same model. Thus, each replication is considered as a sample of a sampling distribution of replications. According to the central limit theorem, the distribution of the replications’ means approximates a normal distribution for a large number of replications (100 in our simulation processes). Based on this assumption, we used the average and the 95% confidence interval’s half-width values of each output measure to estimate the standard deviations.

We tested the assumption of homoscedasticity among the as-is and the to-be models using the F test (α = 0.05), and we used the unequal variance t-test (Welch’s *t*-test; α = 0.05) to compare the mean value of each measure of the as-is and to-be models. The Welch’s *t*-test assumes that the output measures of the compared scenarios are sampled from Gaussian populations of replications and refers to the null hypothesis that the means of the two compared distributions are the same, while the variances may differ [44].

## 3. Results

Table 1 shows the average and 95% confidence interval half-width values of the output measures used to compare the as-is and the to-be scenarios. It also shows the Welch’s t-test *p*-values *(p)* for the cost measures (physiotherapist and healthcare costs next 2 years) and for the other statistically significant variables. 

The increase from 50% to 100% of the patients who use the Dress-KINESIS along the expressed needs pathway does not show statistically significant differences in the comparison between the to-be models and the as-is system. A slight progressive physiotherapist cost saving is detected (from EUR 439.19 to 437.96/month) when the rehabilitation maintenance phase of all patients is managed through the Dress-KINESIS (DK support percentage = 100% in the to-be expressed model).

The added value of using the tool is more evident, and statistically significant, considering the non-expressed needs pathway or the expressed/non-expressed mixed models. In this case, the increased percentage of patients who follow the Dress-KINESIS protocols improves the number of patients in the system at any given time (WIP) and decreases the healthcare costs during the 2 years after pain onset (Table 1). During the 1-month simulation, when the 20% of non-expressed needs are managed through the Dress-KINESIS, the WIP parameter increases from 45.43 (±0.86) to 60.89 (±1.16) patients/month and the 2-year follow-up healthcare costs decrease from EUR 183,662.36 (±2914.46) to EUR 149,210.46 (±2666.38). The same increase of patients in the system at any given time (WIP) is detected when the tool manages the maintenance phase of 80% of expressed needs and assists 20% of the non-expressed needs (Table 1). In this case, the 2-year follow-up healthcare costs decrease to EUR 149,011.38/month (±2617.97).

A higher number of patients in the system (WIP) also determines a slight increase in the average number of patients waiting to see a physiotherapist (from 1.97 to 2.06 patients) but without a statistically significant difference in the waiting time or the average number of physiotherapists in use over time (0.85 ± 0.02).

## 4. Discussion

Healthcare systems all over the world are founded on the historically necessary model of in-person interaction between patients and clinicians. Aging populations and higher chronic disease burdens, even more evident after the COVID-19 pandemic, have invariably raised the pressure on public health systems globally. The waiting time for healthcare services to be delivered has increased worldwide. Patients seeking physical treatment for musculoskeletal impairments have reported that it took up to 70 days to enter a rehabilitation institute. [45].

Consumers’ health behaviors, their attitude towards prevention, their trust in healthcare quality and health costs related to delayed diagnosis or treatment are strongly impacted by the waiting time. Notwithstanding the huge effort by healthcare systems to reduce waiting time, the traditional care delivery model shows structural issues that can no longer be faced only by leveraging human and economic resources. The advancement of digital public health should be considered as a way to reduce waiting times, increase the satisfaction with services and strengthen and personalize offerings of care based on the real status of the patient.

Although some digital technologies have existed for decades, they have had poor penetration into the market because of the reluctance of the medical community, the restrictive and ambiguous regulations governing this innovative space and the sparsity of financial incentives or reimbursement for medical services provided through these technologies [46].

It is worth noting that in 2015, the U.S. Centers for Medicare & Medicaid Services (CMS) issued a final rule on Stage 3 of the Electronic Health Record Incentive Program. This concerns the “coordination of care through patient engagement” which can be performed by involving the use of application program interfaces; view/download/transmit functions; secure messaging; and the ability to upload patient-generated health data (medical device data, home health monitoring data, fitness monitor data, nutritional, home-use medical device data, patient-reported outcome data and so on) into the HER [46]. For public healthcare systems, restrictive or contradictory regulations are the major obstacle to technological advancement, even though many authors showed that the digital model has significant cost advantages, both on the provider side and on the patient side [47,48].

The aim of this work was to use the scenarios methodology to compare the performance of a traditional rehabilitation approach for non-specific LBP patients with an innovative one which involves the use of an mHealth tool, the Dress-KINESIS, in managing different phases of semi-supervised exercise protocols.

Referring to the most recent scientific literature about LBP, we modeled the 1-month behavior of a theoretical model of a rehabilitation institute which faces an LBP point prevalence of 7.5% [31], a rate of low intensity pain patients of 74.5% [41] and a poor prognosis rate of 74% after having performed a first cycle of supervised activity. Four physiotherapists are involved in all of the phases of traditional exercise protocols.

We set the rate of expressed needs to 20% (16% among patients with low-intensity pain and 80% among patients with high-intensity pain) and focused our analysis on the output measures describing the economic impact of non-specific LBP management within the system of care with or without Dress-KINESIS use.

Our results show that an integrated approach of care which involves the use of the Dress-KINESIS could increase the system’s efficiency and its ability to engage patients. In particular, when at least the 20% of non-expressed needs subjects use the Dress-KINESIS, the number of patients in the system increases from 45.43 to 60.89/month, while the physiotherapists’ effort and costs and the waiting time do not change. Additionally, healthcare costs within two years of a new episode of LBP decrease 19%, from EUR 183,662.36 to EUR 149,210.46/month.

These findings support the hypothesis that mHealth technology could promote both primary LBP prevention and early engagement of LBP patients, which are the key strategies in high-risk populations to reduce the burden of chronic diseases and to tackle direct healthcare costs [49].

Other relevant scientific evidence that inspired the development of the Dress-KINESIS should be considered in the scenarios’ evaluation:One-size-fits-all examination and long waiting times could lead patients to feel that they are not being treated as individuals, thus promoting distrust in healthcare services and hesitation to take health assessments [50].Consumers’ spending attitudes are influenced by the cost level of preventive medical services, in particular the out-of-pocket cost, as well as the expected health profit from a medical intervention. This makes health information/education a central point in consumers’ decision-making processes [50].In the real world, up to 70% of patients do not engage in prescribed home exercise, and the lack of time and/or motivation are the main reasons pointed out by people to justify this [51]. This reduces both the adherence to the maintenance activity among patients who have performed a first cycle of supervised exercise and the propensity to perform any structured physical activity among those who do not seek care, in particular when the perceived intensity of pain is low.The most effective treatments for LBP consist of tailored designed exercise programs which are delivered in a supervised format (e.g., home exercise with regular therapist follow-up) [52].Among people who exercise, those who perceive autonomy and receive competent support from exercise experts tend to have greater satisfaction, increased intention to exercise and better adherence [10,51].Individuals who acquire new skills or improve existing ones tend to have greater predispositions and to be more prone to exercise over the long run, especially if their motivation is more self-determined [10,51].

In the real scenario, where public healthcare demands continue to claim an ever-increasing share of public spending, tailored and context-specific solutions are the only way to deliver durable savings and efficiencies in public health systems [53].

A recent metanalysis about the effectiveness of telerehabilitation in physical therapy compared fifty-three systematic reviews of 754 studies performed in Europe, North America and Oceania. The studies mainly concerned the use of the mobile telephone through its messaging services and telephone calls. This work concluded that telerehabilitation could be comparable to or better than the conventional methods of rehabilitation to reduce pain and improve physical function in musculoskeletal conditions generally and, in particular, to improve quality of life in patients with non-specific LBP [54].

Some other studies performed in Kuwait [55], India [56], The Netherlands [57] and Brazil [58] provided evidence of good outcomes for a variety of telerehabilitation interventions to treat musculoskeletal conditions. Notwithstanding these results, comparison among countries and scientific evidence is hard to perform because the term “telerehabilitation” refers to many interventions delivered through a variety of synchronous/real-time (e.g., videoconference) and asynchronous/store-forward (e.g., digital images) means. All of the evidence, however, agrees about (1) the importance of adherence to the ethical principle of causing no harm and (2) the importance of performing patient-centered interventions.

The Dress-KINESIS has been designed to meet these aspects too.

Our most efficient results concern the to-be models, where the demand for care increases, but it is managed through an innovative approach which exploits community collaboration and combined technology offerings. This is in line with the analysis of Ashwood et al., which concluded that the majority of the digital episodes of care represented new utilizations of services [59].

Furthermore, this finding agrees with the emerging trend, not just in digital health, but also in health research in general, of increasing patient engagement by treating the patient as a stakeholder in research, not just a data source [15].

Individual behaviors in health occur under specific conditions and with private information and assessment [50]. Our proposed solution incorporates all of the aspects (greater control of patients of their own health, supervision, exercise protocols tailored on the patients’ abilities and participation in a general behavior change program incorporating motivational strategies) which are known to be complex factors influencing patients’ propensity to adhere to prescribed self-management exercise programs [52]. 

### Strength and Limitations:

This paper shows some limitations. Firstly, the LBP prevalence data and the rehabilitation cost estimates, being drawn from the most recent scientific literature, are characterized by a great variability among geographical areas and health systems’ organizational models. This could have impacted the as-is model development and, as a consequence, the to-be models’ results. Secondly, with respect to the digital care model costs, we underline that the initial investment costs of the Dress-KINESIS start-up phase are not included in our analysis, and this lowers the cost evaluation of our solution. However, as the tool’s use increases, the fixed costs will be spread out over a larger number of users and institutes, and the total costs of implementation and use will reduce over time. This study also shows some relevant strengths. The main advantage is found in the use of the Arena tool to estimate many quantitative parameters (number of treated patients, costs, waiting time, resource use) potentially influenced by the digital tool’s use in different phases of the rehabilitation process for LBP management. The second advantage is found in the technology readiness level of the digital tool, which is currently equal to 4 (basic prototype validated in laboratory environment) on a 9-point scale of the development level.

This allows us to propose the Dress-KINESIS to the general population in Italy during the next 6–8 months and thus further validate the tool itself and our current results about cost saving using real-world patient data.

## 5. Conclusions

The important benefit of the scenario approach lies in its ability to transform “advocacy planning”, which is more prone to expert solutions or professional approaches, into “options planning” by identifying possible sets of future conditions of decision-making utility [2]. Our scenario analysis suggests that citizens’ participation and the use of digital health tools could be the levers of a new paradigm to improve the healthcare system’s efficiency and the citizens’ quality of life.

This is in line with the European agenda for research and innovation [60], which promotes scientific solutions as means of bettering communities’ resilience and suggests engaging European Union citizens in the research and innovation cycle to provide feedback on new technologies and ensure societal uptake of disruptive solutions.

## 6. Patents

The Dress-KINESIS’s logical scheme and brand, as well as the progression criteria along the three cycles of activity, will be protected by CNR through other intellectual property rights.

## Figures and Tables

**Figure 1 ijerph-19-07805-f001:**
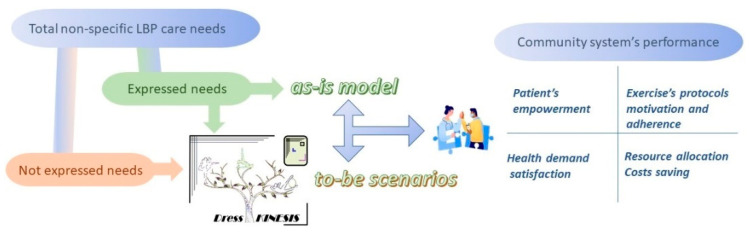
Article topics.

**Figure 2 ijerph-19-07805-f002:**
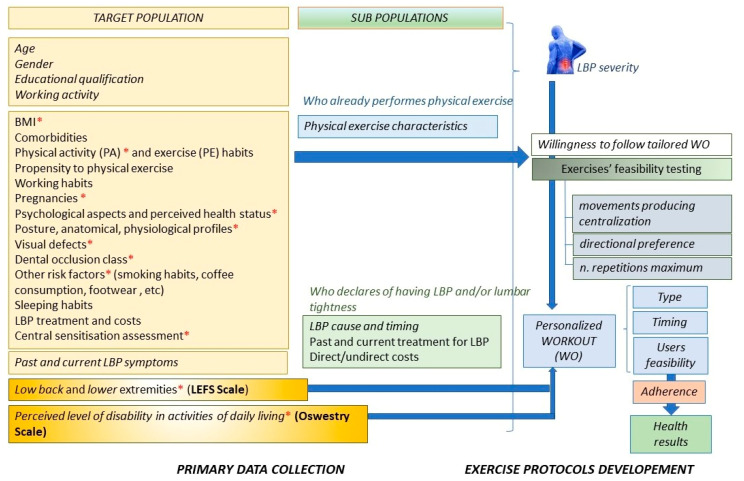
Dress-KINESIS primary data collection and general scheme. The red asterisks indicate the survey items which the educational messages refer to.

**Figure 3 ijerph-19-07805-f003:**
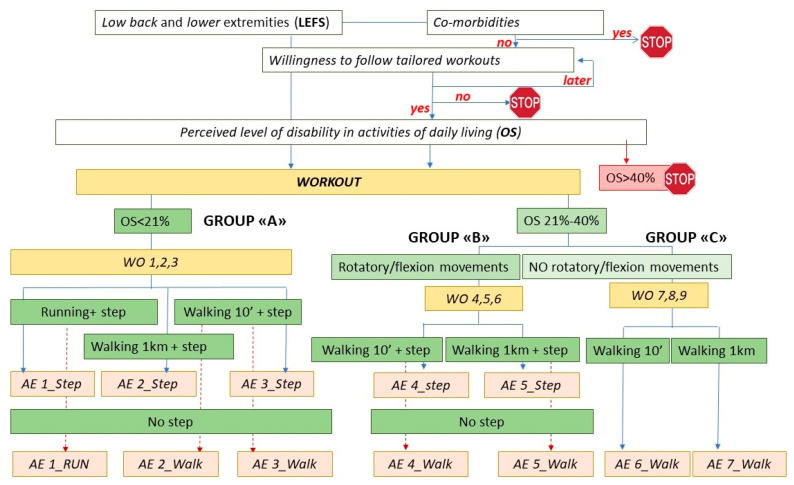
Dress-KINESIS logical scheme.

**Figure 4 ijerph-19-07805-f004:**
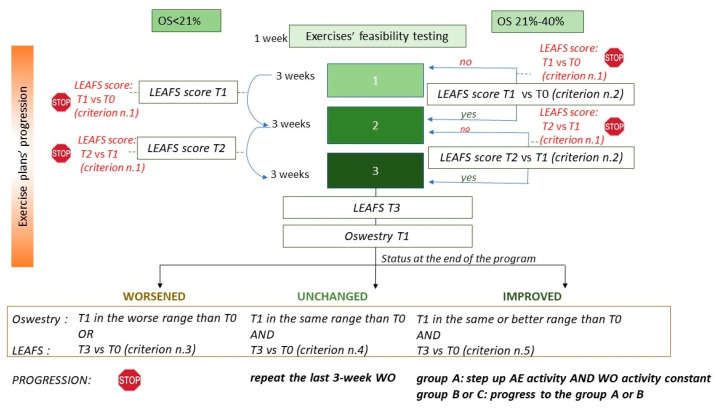
Progression criteria along the three 3-week cycles.

**Figure 5 ijerph-19-07805-f005:**
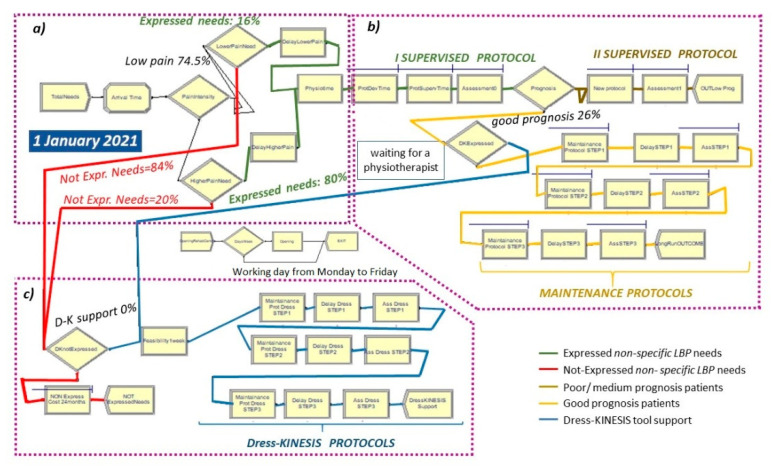
The overall as-is logical model for supporting non-specific LBP patients in conducting targeted physical exercise protocols. Sub-figures (**a**–**c**) highlight the different stages of the process, described in more detail in Figure 6.

**Figure 6 ijerph-19-07805-f006:**
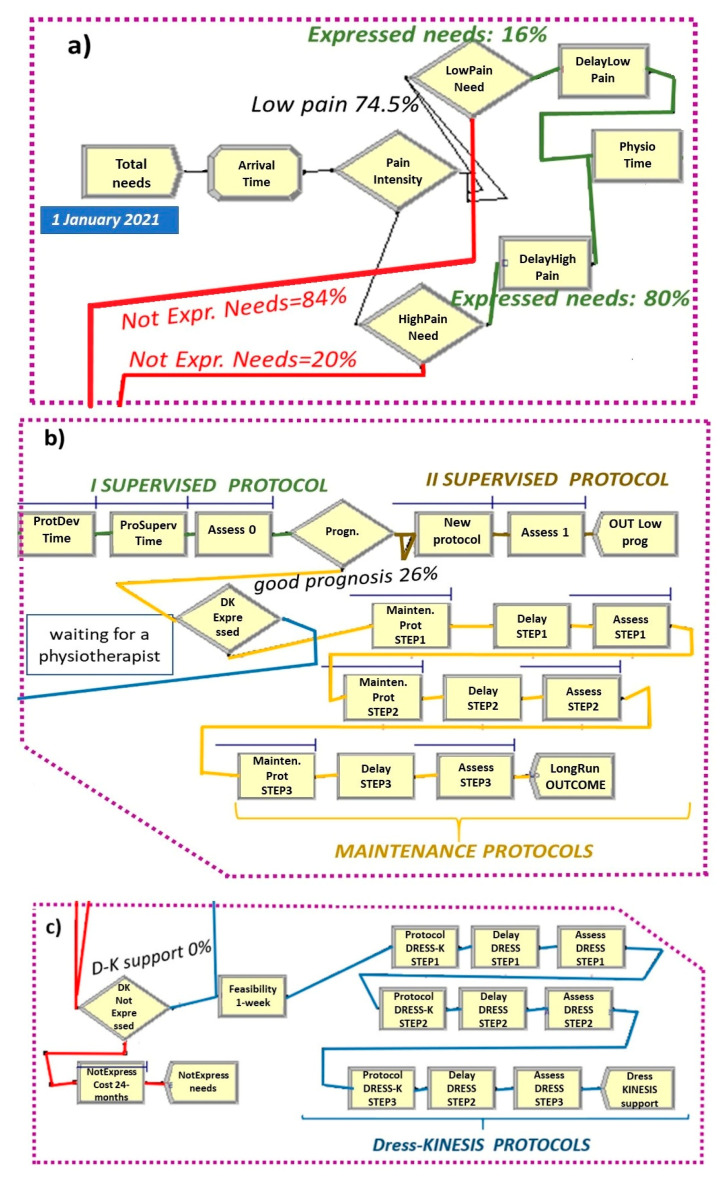
(**a**) Overview of the process stage to differentiate patients by pain intensity and propensity to express their needs of care. (**b**) Overview of the process stage concerning patients who express their needs for care. (**c**) Overview of the process stage concerning patients who do not express their needs of care.

**Table 1 ijerph-19-07805-t001:** Performance measures report and statistical analysis (n. of replications = 100; 1-month activity).

								Physiotherapists		Healthcare Costs Next 2 Years	
Scenario	% +D-K Support	Pat. IN 30 Days (95% CI Half Width)	Av. WIP (95% CI Half Width)	%Δ WIP (p)	Av. Number physio. Use over Time (95% CI Half Width)	Av. Number Patients Waiting for a physio. (95% CI half width)	Average Waiting Days for a Physio. (95% CI Half Width)	Av. Cost (€) (95% CI Half Width)	%Δ Cost (p)	Av. Costs (€) (95% CI Half Width)	%Δ Costs (p)
AS IS	E:0; NE:0	245.52 (±3.09)	45.43 (±0.86)	-	0.85 (±0.02)	1.97 (±0.05)	0.84 (±0.02)	439.10 (±13.47)	-	183,662.36 (±2914.46)	-
TO BE Expr	E:50	245.59 (±3.08)	45.43 (±0.87)	0.0% (>0.05)	0.85 (±0.02)	1.97 (±0.05)	0.84 (±0.02)	438.65 (±13.39)	−0.10% (>0.05)	183,695.54 (±2914.33)	0.02% (>0.05)
	E:60	245.63 (±3.05)	45.43 (±0.87)	0.0% (>0.05)	0.85 (±0.02)	1.97 (±0.05)	0.84 (±0.02)	438.52 (±13.38)	−0.18% (>0.05)	183,717.66 (±2910.66)	0.04% (>0.05)
	E:70	245.66 (±3.05)	45.43 (±0.87)	0.0% (>0.05)	0.85 (±0.02)	1.97 (±0.05)	0.84 (±0.02)	438.31 (±13.38)	−0.18% (>0.05)	183,739.78 (±2910.66)	0.04% (>0.05)
	E:80	245.63 (±3.02)	45.43 (±0.86)	0.0% (>0.05)	0.85 (±0.02)	1.97 (±0.05)	0.84 (±0.02)	438.19 (±13.35)	−0.21% (>0.05)	183,784.02 (±2888.98)	0.07% (>0.05)
	E:100	245.80 (±3.04)	45.43 (±0.86)	0.0% (>0.05)	0.85 (±0.02)	1.97 (±0.05)	0.84 (±0.02)	437.96 (±13.36)	−0.26% (>0.05)	183,994.16 (±2933.87)	0.18% (>0.05)
TO BE Not-Expr	NE:10	246.50 (±3.44)	53.27 (±1.02)	17.26% (<0.05)	0.85 (±0.02)	2.03 (±0.06)	0.87 (±0.02)	439.36 (±13.73)	−0.16% (>0.05)	167,028.12 (±3068.02)	−9.04% (<0.05)
	NE:15	247.32 (±3.26)	57.45 (±1.10)	26.46% (<0.05)	0.86 (±0.02)	2.06 (±0.06)	0.87 (±0.02)	447.36 (±13.71)	1.88% (>0.05)	157,394.86 (±2700.72)	−14.30% (<0.05)
	NE:20	247.11 (±3.64)	60.89 (±1.16)	34.03% (<0.05)	0.85 (±0.02)	2.06 (±0.06)	0.87 (±0.02)	441.66 (±13.90)	0.58% (>0.05)	149,210.46 (±2666.38)	−18.76% (<0.05)
TO BE Mixed	E:80; NE:10	246.50 (±3.48)	53.27 (±1.02)	17.26% (<0.05)	0.85 (±0.02)	2.03 (±0.06)	0.87 (±0.02)	438.41 (±14.52)	−0.16% (>0.05)	167,050.24 (±3062.47)	−9.04% (<0.05)
	E:80; NE:15	247.23 (±3.24)	57.44 (±1.10)	26.44% (<0.05)	0.86 (±0.02)	2.05 (±0.06)	0.87 (±0.02)	445.85 (±13.53)	1.54% (>0.05)	157,350.62 (±2714.94)	−14.33% (<0.05)
	E:80; NE:20	247.02 (±3.59)	60.89 (±1.16)	34.03% (<0.05)	0.85 (±0.02)	2.06 (±0.06)	0.87 (±0.02)	440.48 (±13.69)	0.31% (>0.05)	149,011.38 (±2617.97)	−18.87% (<0.05)

(E: Expressed needs; NE: Non-expressed needs; D-K: Dress-KINESIS; WIP: number of patients in the system at any given time).

## Data Availability

Not applicable.

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
