# Peer review of "Participation in Low Back Pain Management: It Is Time for the To-Be Scenarios in Digital Public Health"

_ijerph, 2022, doi:10.3390/ijerph19137805_

Round 1

Reviewer 1 Report

Over all the paper is well written and discusses an interesting viewpoint.

Some minor changes needed

  • Figures and diagrams are hard to read
  • Minor spell checks and grammatical errors.

Author Response

Dear Reviewer 1,

thank you for your precious comments.

We have made substantial changes to the manuscript according to your indications and to the comments of the other two reviewers. You will find them in the new uploaded manuscript: the addictions and deletions are in track changes mode. As you can see, we also upgraded all the manuscript figures.  In particular, the Figure 5 has been also exploited in 3 subfigures, showing the details of the overall process. 

We have also revised Table 1 to make our results more informative.

Finally, we have revised the English and spell cheks.

We believe these changes have strongly improved our manuscript.

Best regards

Reviewer 2 Report

I was intived to revise the paper entitled "Participation in health: it’s time for the to-be scenarios in public health".  It aimed to evaluate the impact of a web based toll, Dress-KINESIS, in the management of LBP.

I am very skeptics about this paper. It has several criticism.

Mainly, the paper is very confused, with a large methods section hard to read. In addition results are not related to any real data.

In addition:

  • The title is not informative. It does not give any information about the study;
  • Sample size estimation was totally lacking;
  • Authors did not described the enrollment procedure;
  • Authors showed several results but no description of the study population was reported;
  • The statistical analysis proposed should be deeply described and it is not appropriate.

Author Response

Dear Reviewer 2,

thank you for your precious comments.

We have made substantial changes to the manuscript according to your indications and to the comments of the other two reviewers. You will find them in the new uploaded manuscript: the addictions and deletions are in track changes mode. As you can see, we also upgraded all the manuscript figures.  In particular, the Figure 5 has been also exploited in 3 subfigures, showing the details of the overall process. 

We have also revised Table 1 to make our results more informative.

Finally, we have revised the English and spell cheks.

We believe these changes have strongly improved our manuscript.

Moreover, in the atthaced file you will find details in red under your original comments.

Best regards

Reviewer 3 Report

Thank you for your submission on an interesting research topic. While the discussion is important and useful, there are several issues that require your attention and addressing before the paper can be accepted in my general evaluation.

First, the paper will benefit from an early discussion on the background of the health service financing problem. This is critically important to alert readers to the specific details of the paper. Some additional references that the author might wish to refer to include: a) https://pubmed.ncbi.nlm.nih.gov/26413435/; b) https://www.sciencedirect.com/science/article/abs/pii/S1526590006010248; c) https://www.ncbi.nlm.nih.gov/pmc/articles/PMC8438010/

Second, for this research makes sense, it is worthwhile to discuss a better approach to medical innovations (both technical, information elements and business flow), and a recent knowledge management discussion will do justice to the service. One example can be found here (concerning Covid-19 vaccine innovation arrival): https://www.nature.com/articles/s41599-022-01034-6

Third, the paper would need to revamp the quality of graphics. Current graphics do not show very good quality, both resolution and the logical components appearing in each figure.

Fourth, I sincerely believe the audiences will benefit greatly from the authors' careful proofreading and even restructuring of the presentation flow for clarity and readability.

Best regards,

Author Response

Dear Reviewer 3,

We appreciate your valuable suggestions. We have made substantial changes to the manuscript according to your indications and to the comments of the other two reviewers. You will find them in the new uploaded manuscript: the addictions and deletions are in track changes mode. As you can see, we also upgraded all the manuscript figures.  In particular, the Figure 5 has been also exploited in 3 subfigures, showing the details of the overall process. 

We have also revised Table 1 to make our results more informative.

Finally, we have revised the English and spell cheks.

We believe these changes have strongly improved our manuscript.

Moreover, in the attached file you will find details in red under your original comments.

Best regards

Round 2

Reviewer 2 Report

I'm sorry if I have misunderstood the paper but it is obvious that the old version was not easy to read. The methodology was not well written.

After the revision process the paper was substantially improved.

Introduction section is now deep and clearly explaine study background.

The study process is now more easy to read thanks to figures.

I have only some observations:

- The title should be more informative. It should report also the keywords "lowback pain";

- Discussion section remain poor. It should report comparison with previous studies performed for each approach. Authors should discuss also the difference in the management among different countries;

- Strenght and limitation section should be added.

Author Response

Dear Reviewer 2,

thank you again for your precious comments : as a matter of fact, your "misunderstanding" has allowed us to better describe our work.

We have made some improvements in the discussion section (and references integration), according to the other reviewers suggestions, too. We have cited some references concerning the comparison between traditional methods of rehabilitation and telerehabilitation (in particular 46-48 and 50,54,55).

Furthernore, we have added Strenght and limitation section

We have also modify the title in : « Participation in Low Back Pain management : it’s time for the to-be scenarios in digital public health »

We hope to have fully adress your suggestions and that our changes have furtherly improved the manuscript.

Best regards

Reviewer 3 Report

Dear authors,

Thank you very much for your thoughtful revisions. I have seen that the quality has been enhanced significantly along with the paper's readability.

Clearly, the paper has made the case for cost management and m-health. I only have a last minor comment that for the paper to be more convincing, the current lack of discussion on the behavioral economics of healthcare, m-health, health information, and healthcare system sustainability should be addressed. This address will connect various discussions on cost issues smoothly. I introduce a recent paper dealing with the matter with high relevance: https://www.nature.com/articles/s41599-018-0127-3 in hope of aiding this final (minor) revision.

I trust this last thing can be handled efficiently.

Best wishes,

Author Response

Dear Reviewer 3,

thank you again for your further precious suggestion. We have made some improvements in the discussion section (and references integration), according to the other reviewers suggestions, too. We have cited some references concerning the consumers’ health behaviours and the waiting time impact on the consumers’ trust in healthcare quality and health costs. We have also make references to the comparison between traditional methods of rehabilitation and telerehabilitation (in particular see references 46-48 and 50,54,55).

Furthernore, we have added Strenght and limitation section

We have also modified the title in : « Participation in Low Back Pain management : it’s time for the to-be scenarios in digital public health »

We hope to have fully adress your suggestion and that our changes have furtherly improved the manuscript.

Best regards